# Knowledge-Based Design Algorithm for Support Reduction in Material Extrusion Additive Manufacturing

**DOI:** 10.3390/mi13101672

**Published:** 2022-10-04

**Authors:** Jaeseung Ahn, Jaehyeok Doh, Samyeon Kim, Sang-in Park

**Affiliations:** 1Department of Mechatronics Engineering, Incheon National University, Incheon 22012, Korea; 2School of Mechanical and Material Convergence Engineering, Gyeongsang National University, Jinju-si 52725, Gyeongsangnam-do, Korea; 3Department of Mechanical Systems Engineering, Jeonju University, Jeonju-si 55069, Jeollabuk-do, Korea

**Keywords:** additive manufacturing, support reduction, knowledge base, design modification, process optimization

## Abstract

Although additive manufacturing (AM) enables designers to develop products with a high degree of design freedom, the manufacturing constraints of AM restrict design freedom. One of the key manufacturing constraints is the use of support structures for overhang features, which are indispensable in AM processes, but increase material consumption, manufacturing costs, and build time. Therefore, controlling support structure generation is a significant issue in fabricating functional products directly using AM. The goal of this paper is to propose a knowledge-based design algorithm for reducing support structures whilst considering printability and as-printed quality. The proposed method consists of three steps: (1) AM ontology development, for characterizing a target AM process, (2) Surrogate model construction, for quantifying the impact of the AM parameters on as-printed quality, (3) Design and process modification, for reducing support structures and optimizing the AM parameters. The significance of the proposed method is to not only optimize process parameters, but to also control local geometric features for a better surface roughness and build time reduction. To validate the proposed algorithm, case studies with curve-based (1D), surface-based (2D), and volume (3D) models were carried out to prove the reduction of support generation and build time while maintaining surface quality.

## 1. Introduction

Additive Manufacturing (AM) provides a high degree of design freedom by printing parts layer by layer. Since AM is capable of fabricating parts with complex geometries that conventional manufacturing techniques may not be able to manufacture, AM is actively adopted in aerospace [1], medical [2], and automotive industries [3]. Furthermore, AM provides unique capabilities, such as shape complexity, material complexity, hierarchical complexity, and functional complexity [4]. Design for additive manufacturing (DfAM) has been studied to take advantage of these unique capabilities. DfAM aims to maximize product performance by synthesizing these unique capabilities [4] and alleviating the manufacturing constraints of AM.

One of the most considered manufacturing constraints is the printing of a part’s overhang. Due to layer by layer fabrication, a support structure is necessary to print the overhang successfully. However, support structure generation causes an increment in material usage and build time, and it requires postprocessing to remove the support structure. Furthermore, it leads to a low surface roughness of overhangs.

To address this issue, generating fewer support structures on overhangs has been actively studied in the field of DfAM. Since material extrusion-based AM (MEX) machines are a reasonable price to purchase and can handle a variety of AM materials, the MEX process has been considered in previous studies [5,6,7]. Support generation of the MEX process can be affected by a build orientation, an overhang angle between a build plate and design features, and the setting of process parameters. Accordingly, studies for minimizing support structure generation focus in two directions: (1) design modification methods based on an overhang angle to avoid support structure generation, and (2) process parameter control to print overhang features without support structures. First, various design methods are studied to minimize support structure generation. Mirzendehdel and Suresh [8] introduced a topology optimization approach to manage the number of support structures using topology sensitivity. Langelaar [9] developed an AM filter to design self-supporting geometries when optimizing a topology of 2D structures. Fu et al. [10] extended the method to create a three-dimensional part design. The fundamental principle of these studies is to determine a specific overhang angle that does not require support structures, and then to consider the overhang angle in order to avoid support structure generation when performing topology optimization. However, changing the entire part design is inevitable. Once the optimized topology of a part is defined, it is challenging to consider the overhang angle to avoid support structure generation. To deal with this issue, Hu et al. [11] proposed a design method to rotate specific design regions of a part to increase self-supporting design regions that can be printed without the support structure. Although this method effectively alleviates support structure generation, the original part design for AM should be modified to improve manufacturability.

Second, process parameters are controlled to improve the quality of parts without the support structure. Through an experimental study, Pérez et al. [12] investigated that surface roughness, as a quality index, is significantly related to layer thickness and wall thickness. Anitha et al. [13] studied relationships between process parameters, such as build speed, layer thickness, bead width, and the quality of printed parts, such as surface roughness. It was determined that build speed is a predominant process parameter influencing surface roughness. Vyavashare et al. [14] fabricated pyramid-type coupons to identify relationships among process parameters, surface roughness, and dimensional accuracy. Layer thickness, build orientation, and build speed were of concern. They quantified the relationship and then constructed a regression model to optimize process parameters for enhancing surface roughness and reducing build time simultaneously. As a result, layer thickness and build orientation affected surface roughness, while layer thickness and build speed affected geometric accuracy. Vahabil and Rahmati [15] applied both neural network and optimization techniques to develop a model for predicting surface roughness according to process parameter settings. They optimized process parameters to improve quality and minimize build time costs. Ali et al. [16] also determined that layer thickness, bead width, and bead space significantly affect build time; layer thickness and build space are related to material consumption; bead width, raster angle, and bead space are predominant process parameters that affect surface roughness. Jiang et al. [17] determined how the cooling rate of a nozzle, build temperature, build speed, and overhang angle affect surface quality. Surface roughness was high when applying a low overhang angle and rapid build speed, while surface roughness was low at low build temperatures.

A challenging research issue from the literature review is managing support structure generation during the AM process, leading to surface quality deterioration and an increase in build time. Specifically, previous studies did not consider in depth how to avoid support structure generation regarding AM process characteristics, part design, and the quality of printed parts. Since support structures are generated for fabricating overhang features utilizing AM processes, relationships among the AM process, part design, and the quality of printed parts should be considered to ensure a quality and successful fabrication by AM. Furthermore, the design region with overhang features should be modified to minimize design changes instead of changing the entire part design, as performed in topology optimization-based design modification.

A knowledge-based design algorithm is introduced to address these issues by reducing support structure generation by local geometry modification and process parameter optimization. The proposed method supports engineers in searching predominant parameters of the MEX process related to the quality, optimizing the dominant parameters through parametric study, and then modifying geometries that require the support structure to minimize support generation. It has three main steps, as shown in Figure 1: knowledge base development, parametric studies with parameters of the MEX process, and then the support reduction algorithm.

This paper consists of the following sections: Section 2 describes an overview of the proposed method. Section 3 introduces knowledge base development and its application for guiding how to select parameters related to quality. Section 3 also depicts a parametric study for optimizing parameters as an application of the knowledge base. Section 4 explains the support reduction algorithm for minimizing support structure generation. Section 5 provides design examples to validate the proposed algorithm. Finally, Section 6 concludes this paper.

## 2. Overview of Knowledge Base Design Algorithm

The proposed method consists of 3 steps: AM ontology construction, parametric study and surrogate modeling, and support reduction algorithm development. It aims to automate design modifications for less support generation by optimizing the process and local geometry modification algorithm. Figure 1 describes the overview of the proposed method.

First, a knowledge base is developed based on an ontology scheme to store the relationships among design parameters, process parameters, and the quality, such as sagging failure and surface roughness of the MEX printing process. The knowledge base, called AM ontology here, helps users to search parameters related to the quality of printed parts.

Second, the parametric study and statistical analysis are performed to quantify the relationships with experiments, to determine the optimal value of process parameters for better quality, such as surface roughness. Predominant process parameters of the MEX process relating to surface roughness are selected by AM ontology and description logic (DL) query language, which is one of the query languages used for searching for information in the ontology. Then, experiments are performed to discover which parameters affect the surface roughness of the printed part and which are the most essential factors relating to quality. Furthermore, a surrogate model is constructed by analyzing experimental data to predict the surface roughness of the printed parts when the process parameters are decided.

Third, local geometry modification is introduced to detect the geometries of parts which require support structure, named as overhang feature, and then to modify the overhanging feature to avoid support structure generation. Furthermore, process parameters are optimized to provide a better quality. The optimized values of the process parameters, such as overhang angle and deposition speed from the second step, are used as the threshold for detecting local geometries for modification. We performed case studies to assess the proposed method. The proposed design algorithm has been applied to curve-based, surface-based, and general 3D volume models to minimize support structure generation. To summarize, the proposed method is used to decide whether support structures in design regions with overhangs are required. If the design regions with an overhang have a good surface roughness without a support structure, then the process parameters are considered optimized.

## 3. Knowledge Base Construction

### 3.1. AM Ontology Development

The AM knowledge base is developed to store information on the manufacturing constraints of AM, which can be translated into the relationships among geometry parameters, process parameters, and the quality of the final parts, such as surface roughness. Although the relationships have been studied previously, the representation has not been formalized to be shared with domain experts. Accordingly, the AM ontology is required to formalize and store the complex relationship information so that the domain knowledge can be reused. Furthermore, it can be used to search for parameters which are related to the quality of the final parts. The searched parameters are used as basic information when planning experiments in Section 3. If new AM technology is considered, such as powder bed fusion and direct energy deposition, AM ontology can embrace these AM processes by considering process parameters and materials as classes of AM ontology.

AM ontology in this study is formalized by Web Ontology Language (OWL). OWL has been applied to represent newly updated information on DfAM with a formal structure, and to help designers search for practical information by reusing it. Furthermore, it is to provide essential information for the design of the experiments, as mentioned before. Ontology has been used in the field of AM to support designers. Kim et al. [18] proposed AM ontology to store various design rules and apply them to analyze the manufacturability of parts. Sanfilippo et al. [19] used ontology to manage AM knowledge and to support designers’ decision-making in a computer system.

Since OWL consists of terminology and correlations among entities, OWL can represent domain knowledge that is a kind of database with text-based information. The Protégé tool is used to build the knowledge base [20,21]. The proposed AM ontology is developed on top of the knowledge base for DfAM [15,18], emphasizing relationships between MEX process parameters, physics, and the quality of final parts, as shown in Figure 2. Geometry, process, and physics are closely related to support structure generation and are critical factors which influence the quality of the final product printed by AM. These relationships are crucial information for developing a knowledge base that can infer knowledge that users want to know. Information on the relationships in the MEX process was collected from previous studies [22,23,24].

Therefore, the AM ontology is developed with four main classes: machine, model, parameter, and physics, as shown in Figure 3. ‘Machine’ class has subclasses with each part of the MEX machine, e.g., bed plate, nozzle, and motor. ‘Model’ class represents the quality of final parts and manufacturing failures, such as sagging and warping failure. ‘Parameter’ class consists of process, geometry, and material parameters. Overhang angle and layer thickness are subclasses of the geometry parameters, while nozzle speed, nozzle temperature, and fan speed, to name but a few, are listed as subclasses of the ‘parameter’ class. The ‘parameter’ class has three parameters to describe design rules extracted from manufacturing constraints [18]. Design rules are relationships among process, material, and geometry parameters, such that the ‘parameter’ class in the AM ontology aims to represent those design parameters which can be printed by the selected process with the given material. ‘Physics’ class is newly considered to represent and search the process parameters of MEX which affect the surface roughness of the final part. Since material extrusion (MEX)-type additive manufacturing (AM) melts plastic materials by heat, process parameters of ‘parameter’ class are selected based on governing equations relating to injection molding and thermal deformation. The defined governing equations are described in Table 1. The determined process parameters from the governing equations are listed in Table 1.

The knowledge base developed by ontology consists of three elements for representing semantic information: subject–predicate–object. Subject and objects are classes of the AM ontology, while properties are relationships between classes, such as ‘subclass of’ and ‘impact on’. When developing the proposed AM ontology, information in Table 2 is applied. The combination of the subject–predicate–object structure in Table 2 is visualized as a graph network, as shown in Figure 4. Nodes are classes, which are subject or object, and edges are object properties that are ‘impact on’ and ‘subclass of’, and are predicate. Figure 4a represents that ‘Nozzle Diameter’, ‘Injection Pressure’, ‘Viscosity’, and ‘Nozzle Speed’ impact ‘Rate of Extrusion Volume’. ‘Rate of Extrusion Volume’ also impacts ‘Extrusion Failure’. Figure 4b represents the hierarchical relationships of the ‘AM parameter’. The AM ontology with this structure can infer new information from existing data and develop a description logic (DL) query for searching for desired information.

### 3.2. Identification of Predominant Parameters by AM Ontology

Since classes and subclasses are intricately interwoven in the developed AM ontology, it is challenging to search for necessary information. In this study, a description logic (DL) query is applied, a query language for searching classes based on Manchester OWL syntax supported by the plugin of Protégé software [20]. Information in Table 2 that is used to build the AM ontology is necessary for query retrieval by DL query. An example of a DL query is shown in Figure 5. The question, “which process and geometry parameters have an impact on sagging failure?”, is represented as DL query “impact_on some Sagging_Failure and (Process_Parameters or Geometry_Parameters)”. The query result is ‘Build_Angle’, ‘Effective_Area’, ‘Layer_Thickness’, and ‘Nozzle_Speed’. The AM ontology and DL query determine process parameters related to surface roughness. The selected process parameters are optimized to manage the surface roughness by performing a parametric study that is described in the next step.

### 3.3. Materials and Methods of Parametric Study for Selected Parameters

The parametric study consists of three steps. The first step is the selection of predominant process parameters relating to surface roughness. From the proposed AM ontology, we find two geometric and three manufacturing parameters: overhang angle, effective area, layer thickness, nozzle temperature, and deposition speed. These parameters are selected because they affect the printed surface quality and roughness. The second step is the design of experiments to prepare for the parametric study. The selected parameters from the first step are used as the baseline information. We have evaluated and selected the process parameters that lead to significant increases in build time: overhang angle, layer thickness, and deposition speed. Then, a mixed-level full factorial design approach is applied to set up the parametric study with the levels for each parameter, as listed in Table 3. The third step is to print bar-shaped coupons by changing the combination of three process parameters. The bar-shaped coupon with dimensions 6 × 10 × 35 mm in Figure 6a is designed, and 90 coupons (three coupons for each combination) are printed with a polylactic acid (PLA) filament from Esun^®^ in an in-house MEX printer in Figure 6b, with preset process parameters in Table 4. Roughness on the downward surface, denoted in Figure 6a of printed coupons, is measured using the Dektak^®^ XT-E stylus profiler from Bruker in Figure 6c. The roughness of coupons is measured once at the center of the surface.

### 3.4. Interpretation of Parametric Study and Constuction of a Surrogate Model

This section presents the results of the parametric study. Table 5 lists the mean value and standard variation of measured surface roughness for each combination of parameter settings, and Figure 7 shows the effect of the selected parameters from the AM ontology. From the results, two trends need to be addressed. Firstly, the surface roughness decreases as the overhang angle increases, regardless of layer thickness and deposition speed. This trend corresponds to the common support-structure generation approach of conventional slicing software, which adds support structures at low overhang angle regions. Next, the deposition speed impacts the surface roughness when a thin layer is applied, but its effect is limited when thick layer is applied. This trend means that a discretization error, such as a stair step originating from the slicing procedure, becomes a dominant factor for surface roughness of as-printed parts when the thicker layer is applied. However, when a thin layer is applied, printed surface roughness is more sensitive to other parameters, the deposition speed, and overhang angle. With 0.1 mm layer thickness, the faster deposition speed and the lower overhang angle lead to a worse surface quality.

A one-way analysis of variance (ANOVA) is performed to quantitatively assess the effect of parameters. Three parameters, overhang angle, layer thickness, and deposition speed, are investigated to confirm which parameter is statistically meaningful for surface quality. Based on the above qualitative examination, four cases are tested considering the interaction between the layer thickness and deposition speed, as listed in Table 6. Table 6 shows a list of *p*-values for four cases. The *p*-value represents significance probability, and a small *p*-value means the parameter affects surface roughness. We set a criterion for *p*-value to 0.05 for a 95% of significance level, to select statistically meaningful parameters. Two parameters are selected from ANOVA tests as affecting parameters to surface roughness, the overhang angle, and deposition speed with a 0.1 mm layer thickness.

Based on the selected parameters, a surrogate model was constructed to predict surface roughness for a given parameter value, as follows:R_a_(A, S) = 26.03 − 0.27 A + 3.08 S − 0.07 A S(1)
where R_a_ represents an expected surface roughness, and A and S are the overhang angle and deposition speed, respectively. The obtained prediction model is utilized in the proposed support reduction algorithm in the next section.

## 4. Support Reduction Algorithm Development

This section explains a support reduction algorithm utilizing the proposed AM knowledge base. The proposed algorithm consists of two steps. In the first step, an input geometry is modified to remove regions where the overhang angle is less than a threshold angle. Here, we set the threshold angle to 20 degrees based on the parametric study result, because we cannot obtain a better surface roughness of less than 32.3 μm. We utilize an optimization scheme for local geometric features to avoid global shape change. In the second step, optimized deposition speed, depending on an overhang angle at a specific location, is determined to improve as-printed surface roughness. The overall procedure is shown in Figure 8.

### 4.1. Shape Modification

In this research, an input geometry was locally modified to remove low overhang angle regions less than the threshold overhang angle because the MEX process does not guarantee good, as-printed surface quality in the regions. To keep an overall shape similar, we formulated a shape optimization problem to minimize the change in shape. To formulate the optimization problem, bounding surfaces of interested regions were modeled using Bezier surfaces so that the bounding surfaces can be easily modified by changing the location of control vertices. The optimization problem is formulated as the following:(2)$$\begin{array}{cc} Find & x\\ Minize & f\left(x\right)=\frac{1}{n}\sum_{i=1}^{n}\sqrt{P_{0}^{i}-P^{i}\left(x\right)}\\ Subject\;to & g\left(x\right)=A_{T}-min\left(A^{i}\left(x\right)\right)<0 \end{array}$$
where, x is a vector containing coordinates of control vertices, n is the number of sampled points on the surfaces, $P_{0}^{i}$ is the location of a sampled point on the original surfaces, and $P^{i}\left(x\right)$ is the location of a sampled point on the modified surfaces. $A_{T}$ is a threshold angle set to 20 degrees for modification and $A^{i}\left(x\right)$ is an overhang angle at a sampled point on the modified surfaces. The objective function, f(x), represents an average change in shape, and the function, the constraint function g(x), is set to keep the minimum overhang angle in a modified geometry larger than the threshold angle. The formulated optimization problem is solved in MATLAB. The schematic procedure for shape modification is shown in Figure 9.

### 4.2. AM Process Modification

Process parameters are modified based on an optimization scheme to guarantee the as-printed surface quality without support structures. We focused on obtaining an optimized deposition speed because the overhang angle and layer thickness are fixed during fabrication. The optimized deposition speed is determined by minimizing the difference between the expected as-fabricated surface roughness from Equation (1) and the numerically determined surface roughness, as shown in Figure 10. To find the optimized deposition speed, we formulated an optimization problem as follows:(3)$$\begin{array}{cc} \mathrm{Find} & S\\ \mathrm{Minimize} & {\left(R_{a}^{\mathrm{CAD}}-R_{a}\left(S\right)\right)}^{2}\\ \mathrm{Subject}\;\mathrm{to} & 10<S<50 \end{array}$$
where, $R_{a}^{CAD}$ is the numerically determined surface roughness from a modified CAD and $R_{a}\left(S\right)$ is the estimated surface roughness for a given overhang angle. The optimized deposition speed from Equation (3) is exported to a g-code file with slicing information of a modified geometry.

## 5. Result of Case Studies and Discussion

The proposed support reduction approach was implemented in three parts for the case study. Printing material and conditions were the same as those for the parametric study in Section 3.3. The same surface profiler was used to measure the surface roughness. The surface roughness and build time were analyzed in quantitative and qualitative manners.

### 5.1. Curve-Based Model

The proposed approach was applied to a curve-based geometry shown in Figure 11a, which was generated by extruding a set of curves. In this part, the same shape and process parameter modification can be applied through its cross-section because this part has a constant cross-section along the extruding direction. This part has two overhang regions, as shown in Figure 11b, and local shape modification was performed to remove the regions. The modified cross-section of the part is shown with its original cross-section in Figure 11c. The proposed shape modification approach locally adjusts geometries near overhang regions, so that the overall shapes of the part remain the same. Overhang angles in the corrected geometry become larger than the threshold angle after modification. However, the overhang angles between 20 and 40 degrees, which yield rough surfaces, are still in the part geometry.

In the next step, optimization was performed to find the deposition speed that led to better surface roughness. The deposition speed was determined for each layer, and the same speed was applied to each layer. Figure 12 compares the resulting as-fabricated surfaces using three slicing approaches without support structures: (a) an original shape with a conventional slicer, (b) an original shape with an in-house slicer utilizing only process modification, and (c) a modified shape with an in-house slicer utilizing the proposed approach. As expected, the proposed approach yielded the smoothest surface with a roughness of 23.10 μm.

### 5.2. Surface-Based Model

The following example is a geometry that has a freeform surface, shown in Figure 13a. In contrast to the part in the previous section, the shape and process modification is applied pointwise. The input geometry is modified based on two threshold overhang angles, 20 and 45 degrees, to compare and investigate effectiveness of the proposed approach. Figure 13b,c show modified shapes for two threshold angles. When the threshold angle is set to 20 degrees, the shape modification has occurred in a relatively small region. The shape modification with 45 degrees leads to a self-supported shape, which does not require support structure in conventional slicing software. Still, its geometrical characteristics, including normal vectors and curvature, are widely changed.

We fabricated five cases using three geometries, including an original model and modified models with 20 degrees and 45 degrees. The original geometry was used without process modification in the first three cases. Support structures were added in Case 1 but not in Cases 2 and 3. Process modification was applied in Case 3 but not in Case 2. For Case 4, the modified model with the threshold angle of 20 degrees was used, but process modification was applied. For Case 5, the modified model with the threshold angle of 45 degrees was printed without support structure or process modification. Table 7 compares the cases and build time for fabrication, and Figure 14 shows as-printed surfaces for each case. As expected, build time for cases without support structures was reduced by up to 70%. It takes a similar build time for Cases 2–5. However, smooth surfaces with low roughness were observed only in Cases 4 and 5. It is noted that the geometry of Case 5 was self-supporting, but that of Case 4 was not self-supporting because of regions with overhang angles below 45 degrees.

### 5.3. Three-Dimensional Model

To validate the applicability of the proposed approach, we implemented the proposed algorithm to a 3D model of a Moai statue, as in Figure 15a. The shape modification process was not applied because this model had small overhang regions below 20 degrees. However, process modification for the deposition speed was performed in overhang regions below 45 degrees to improve the surface quality. Figure 15b shows the optimized deposition speed mapping result. The deposition speed for the regions below 20 degrees of overhang angle was set to 1 mm/s, and the speed was increased in higher overhang angle regions. Figure 16 compares as-printed surfaces utilizing conventional slicing software (Cura) and the proposed approach. The presented algorithm resulted in a better surface quality without sagging material on surfaces.

## 6. Conclusions

Support structure generation is inevitable for part fabrication by AM due to the layer by layer manners of AM. However, support structure generation affects the qualities of printed parts by MEX and makes the parts inapplicable if supports cannot be removed, such as the part with the conformal cooling channel with supports. Therefore, the proposed method was developed to avoid support structure generation with two essential techniques: process parameter optimization to ensure surface roughness without support structure on the overhang feature, and local geometry modification to prevent support generation. The contribution and novelty of the proposed methods are described below.

Knowledge base development to formalize and store relations among geometry, process, and qualityKnowledge base application to retrieve predominant process parameters related to surface roughnessAutomatic design modification by selecting and redesigning specific overhang regions to avoid support structure generation and to ensure surface quality, which is an objective function of the support reduction algorithmProcess optimization to prevent support structure generation of the selected specific overhang regions

A case study with curve-based, surface-based, and 3D models with various overhang features was performed to evaluate the proposed method—the presented method results in significantly enhanced surface quality and reduced build time. Since support structure generation is related to surface quality, build time and cost, and support removal, automatically detecting overhang regions and preventing support generation by the proposed method is significant. A limitation of this study is that design modification may lead to changes in mechanical performance. Accordingly, future studies will consider how design modification affects the quality of final parts, including mechanical performance and dimensional accuracy, which were not considered in this study. Furthermore, more indexes relating to the quality of the printed part will be regarded as fabricating parts with satisfactory quality. The proposed method will be applied to parts with more complex geometries for further evaluation.

## Figures and Tables

**Figure 1 micromachines-13-01672-f001:**
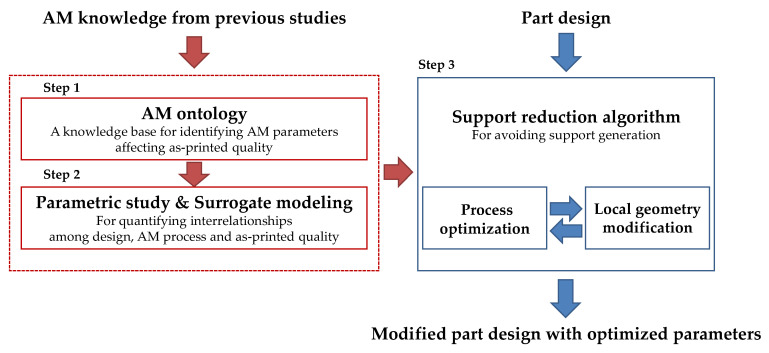
Overview of the proposed method.

**Figure 2 micromachines-13-01672-f002:**
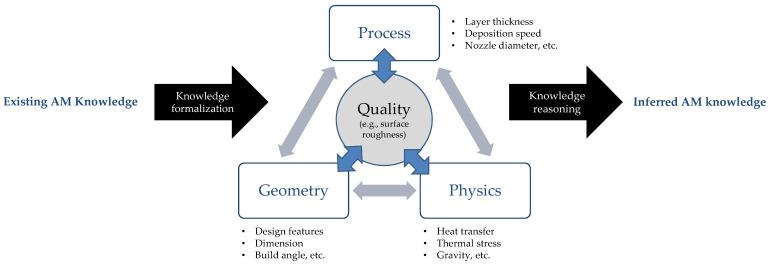
Relationships among process, physics, geometry, and surface roughness.

**Figure 3 micromachines-13-01672-f003:**
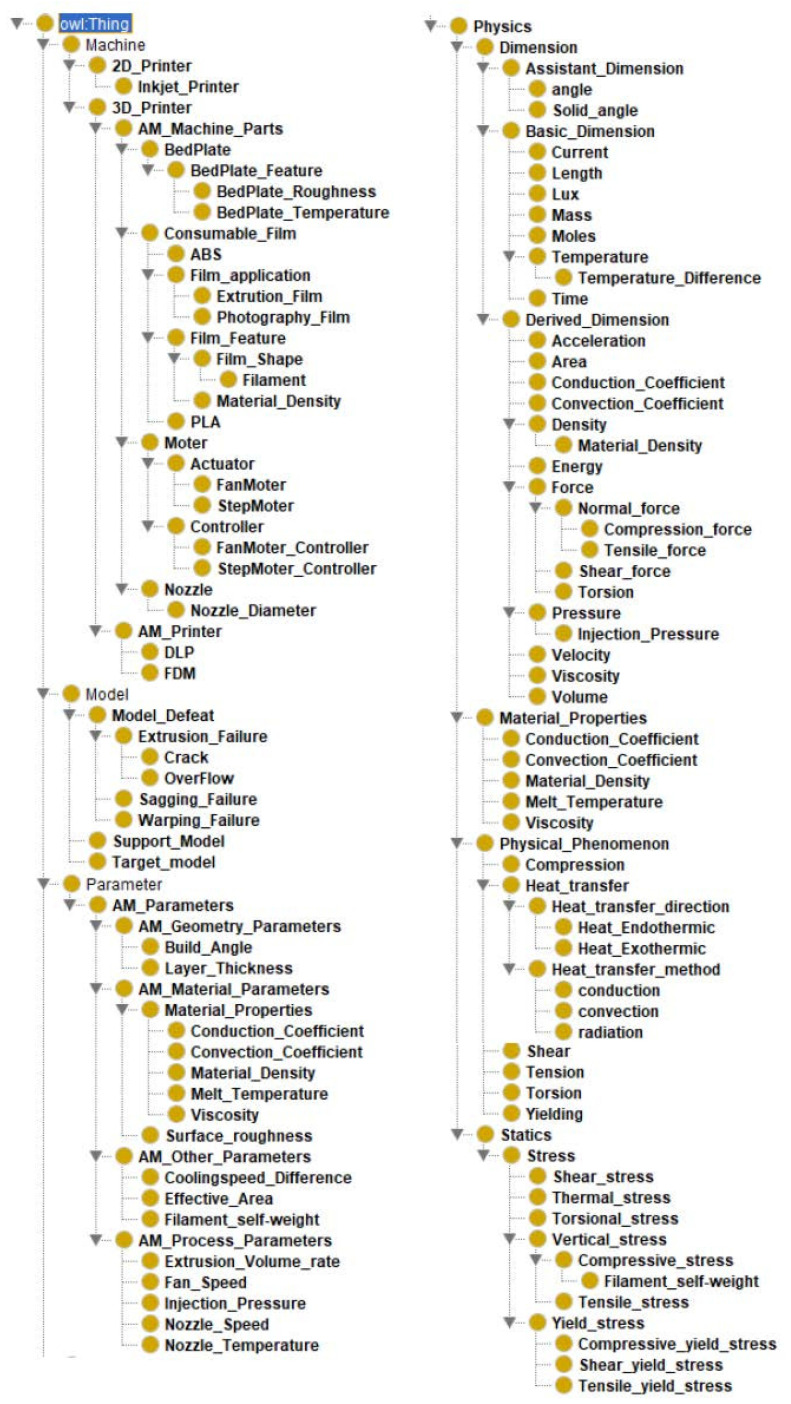
Structure of developed AM ontology in Protégé software (version 5.5.0).

**Figure 4 micromachines-13-01672-f004:**
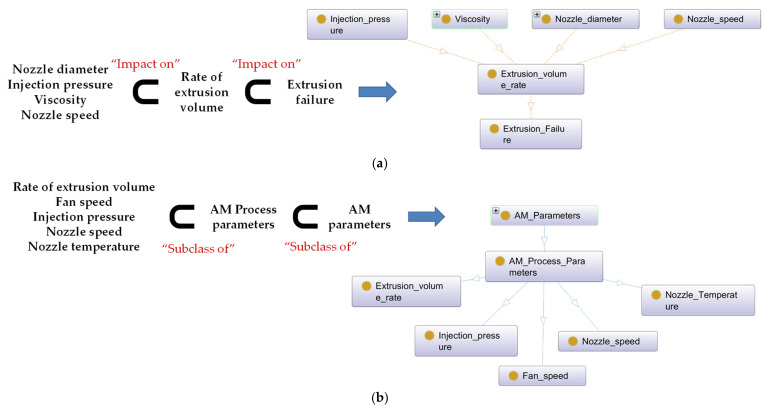
Schematic diagram for constructing AM ontology: (**a**) Implementation of “Impact on”; (**b**) Implementation of “Subclass of” in Protégé software.

**Figure 5 micromachines-13-01672-f005:**
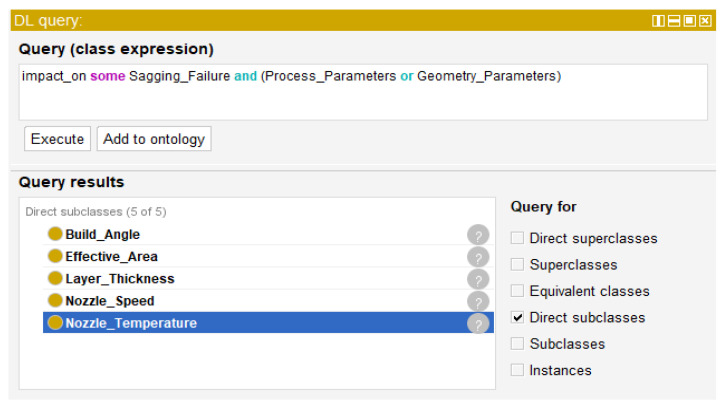
An example of search results by DL query in Protégé software.

**Figure 6 micromachines-13-01672-f006:**
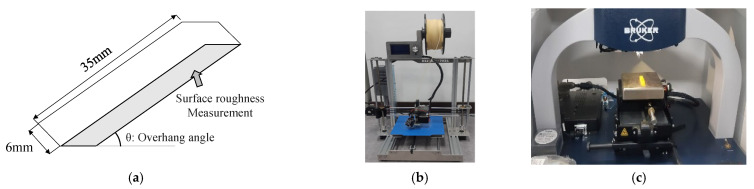
Test preparation: (**a**) Coupon configuration; (**b**) in-house MEX printer; (**c**) Surface profiler.

**Figure 7 micromachines-13-01672-f007:**
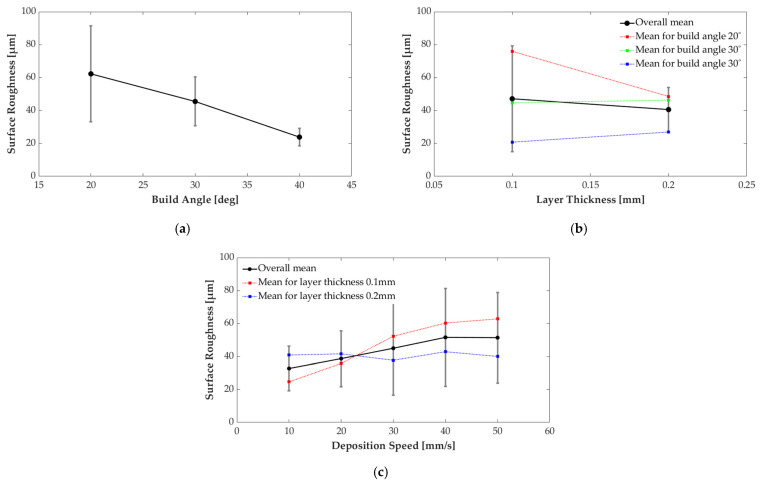
Effect of selected process parameters on surface roughness: (**a**) overhang angle; (**b**) Layer thickness; (**b**) Deposition speed.

**Figure 8 micromachines-13-01672-f008:**
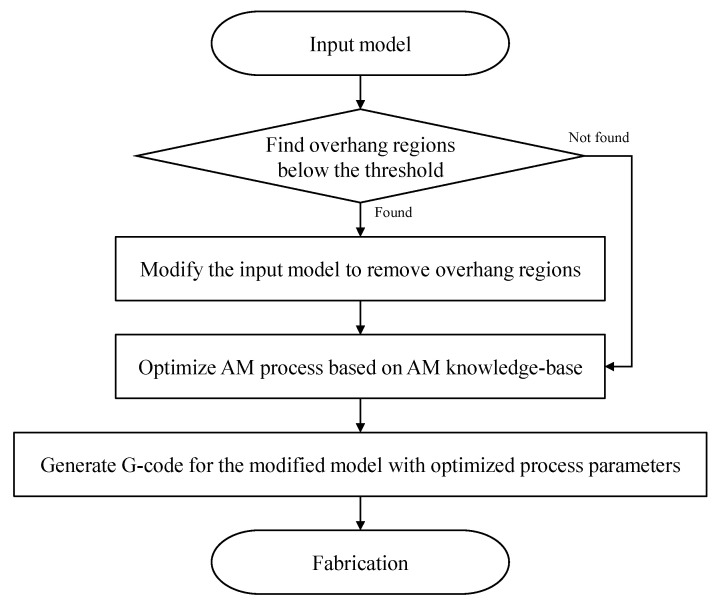
An example of search results by DL query.

**Figure 9 micromachines-13-01672-f009:**
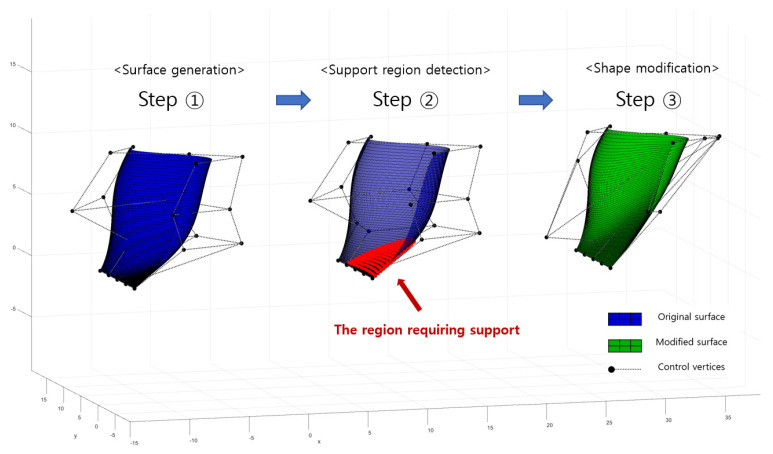
Schematic procedure for local shape modification.

**Figure 10 micromachines-13-01672-f010:**
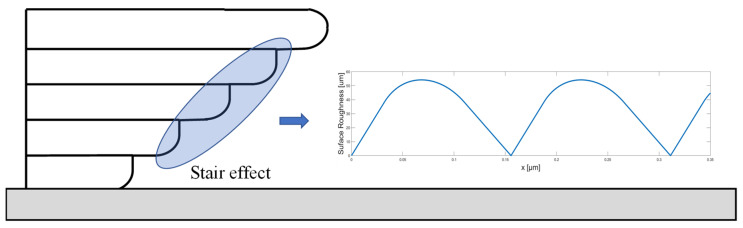
Numerical estimation of surface roughness.

**Figure 11 micromachines-13-01672-f011:**
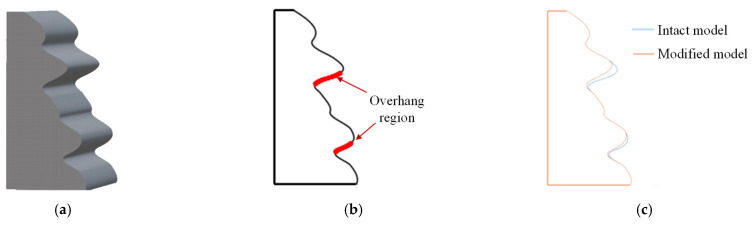
Curve-based model: (**a**) 3D model; (**b**) Base curve; (**c**) Modified curve.

**Figure 12 micromachines-13-01672-f012:**
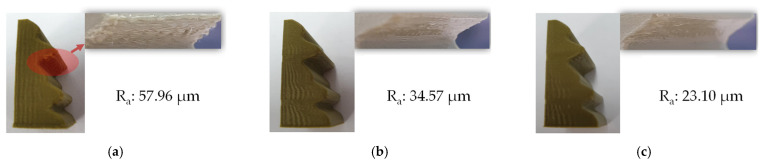
Comparison of fabricated parts: (**a**) Original shape with conventional slicer, (**b**) Original shape with process optimization only (in-house slicer), (**c**) Original model with the proposed approach (in-house slicer).

**Figure 13 micromachines-13-01672-f013:**
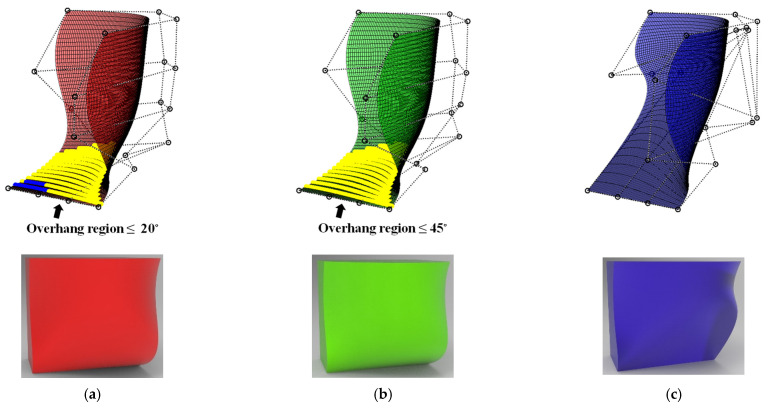
Comparison of surface-based models: (**a**) Original model; (**b**) Modified model with 20° threshold angle; (**c**) Modified model with 45° threshold angle.

**Figure 14 micromachines-13-01672-f014:**
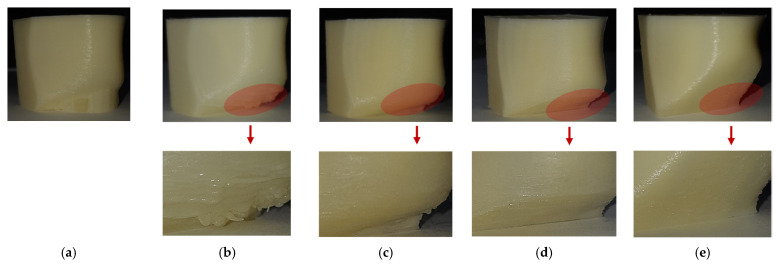
Comparison of as-fabricated models: (**a**) Case 1, (**b**) Case 2, (**c**) Case 3, (**d**) Case 4, (**e**) Case 5.

**Figure 15 micromachines-13-01672-f015:**
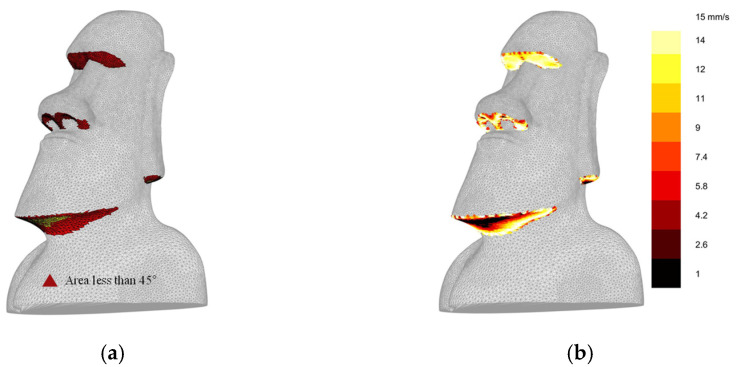
Characteristics of the example model: (**a**) Overhang region, (**b**) Optimized deposition speed map.

**Figure 16 micromachines-13-01672-f016:**
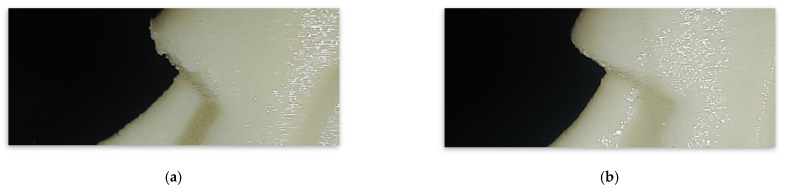
Comparison of as-fabricated surfaces: (**a**) Using conventional slicing software; (**b**) Using in-house slicer with the proposed algorithm.

**Table 1 micromachines-13-01672-t001:** Process parameter identification from governing equations of physical phenomenon in AM.

Governing Equations		Related AM Process Parameters
Viscous flow	$Q=\frac{\pi r^{4}△P}{8L\mu }$	Pressure, Viscosity, Nozzle diameter, Flow rate
Thermal deformation	$\epsilon_{T}=\alpha △T$	Coefficient of thermal expansion, Temperature difference
Stress	$\sigma =\frac{F}{A}\;$	Force, Stress, Area
Heat convection	$Q_{conv}=hA△T\;$	Coefficient of heat convection, Temperature difference, Surface temperature, Area
Heat conduction	$Q_{cond}=kA\frac{△T}{△x}$	Coefficient of thermal conduction,950722Temperature difference, Area

**Table 2 micromachines-13-01672-t002:** Example for elements of knowledge-base.

Class (Subject)	Object Property (Predicate)	Class (Object)
Nozzle Speed	impactOn	Extrusion Flow
Nozzle Diameter	impactOn	Extrusion Flow
Injection Pressure	impactOn	Extrusion Flow
Extrusion Flow	impactOn	Extrusion Failure
Viscosity	impactOn	Extrusion Flow
Nozzle Temperature	impactOn	Viscosity
Nozzle Temperature	subClassOf	Process Parameter
Shear stress	impactOn	Sagging Failure
Effective Area	impactOn	Shear stress
Overhang angle	impactOn	Effective Area
Overhang angle	subClassOf	Geometric Parameter
Cooling rate	impactOn	Warping Failure
Fan Speed	impactOn	Cooling rate
Fan Speed	subClassOf	Process Parameter
Sagging	need	support
Warping	need	support

**Table 3 micromachines-13-01672-t003:** Level of process parameters for parametric study.

Process Parameters	Level				
	1	2	3	4	5
Overhang angle (deg)	20	30	40		
Layer thickness (mm)	0.1	0.2			
Deposition speed (mm/s)	10	20	30	40	50

**Table 4 micromachines-13-01672-t004:** Preset process parameters for coupon fabrication.

Preset Process Parameters	Value
Nozzle temperature (°C)	190
Bed temperature (°C)	50
Infill density (%)	40

**Table 5 micromachines-13-01672-t005:** Mean value (standard deviation) of measured surface roughness.

Layer Thickness (mm)	Overhang Angle (Degree)	Deposition Speed (mm/s)				
		10	20	30	40	50
0.1	20	32.3 (7.3)	61.4 (5.1)	92.2 (11.2)	92.0 (13.7)	102.0 (10.5)
	30	24.1 (1.4)	23.9 (6.1)	49.4 (14.5)	63.6 (4.0)	58.3 (1.3)
	40	17.5 (1.1)	20.4 (0.4)	22.0 (0.3)	23.1 (1.4)	25.6 (0.6)
0.2	20	45.6 (13.1)	55.1 (10.5)	39.9 (7.9)	51.9 (16.2)	50.2 (14.4)
	30	49.7 (1.1)	42.5 (3.1)	46.0 (3.9)	50.7 (6.4)	42.6 (10.4)
	40	29.3 (5.4)	24.9 (2.0)	22.7 (2.2)	22.9 (4.2)	23.7 (3.8)

**Table 6 micromachines-13-01672-t006:** Result of one-way ANOVA.

Process Parameters		Value
Overhang angle (A)		1.02 × 10^−10^ < 0.05
Layer thickness (L)		0.158 > 0.05
Deposition speed (S)	With 0.1 mm of layer thickness	0.042 < 0.05
	With 0.2 mm of layer thickness	0.951 > 0.05

**Table 7 micromachines-13-01672-t007:** Comparison of fabrication cases.

Case	Geometry	Support Structure	Process Modification	Build Time (Second)
1	Original	○	×	3580
2	Original	×	×	2607
3	Original	×	○	2598
4	Modified (≥20°)	×	○	2618
5	Modified (≥45°)	Not required	Not required	2535

## Data Availability

Not applicable.

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
