# Peer review of "Knowledge-Based Design Algorithm for Support Reduction in Material Extrusion Additive Manufacturing"

_micromachines, 2022, doi:10.3390/mi13101672_

Round 1

Reviewer 1 Report

This research proposed a knowledge based algorithm to reduce the support and improve the surface for FDM process. Here are some questions for the authors:

1. Is this algorithm limited to FDM? If it is used for other AM processes, what changes are needed?

2.  The model is modified to reduce the overhang region, like figure 11, but what if in some cases it will change the mechanical performance of the model?

3. There are some typos in this manuscript, like section 3.3, "surrofate". Also there are a lot of sources not found in this document. The authors should proofread the manuscript to modify the typos and errors.

Author Response

We appreciate your valuable comments on this manuscript. We have revised our manuscript to reflect the issues you pointed out. As a result, we believe the manuscript becomes more convincing with revision. We list our point-by-point answers to your questions in this rebuttal letter and have updated the manuscript based on the answers.

(Comment) This research proposed a knowledge-based algorithm to reduce the support and improve the surface for FDM process. Here are some questions for the authors:

  1. Is this algorithm limited to FDM? If it is used for other AM processes, what changes are needed?
  • Thank you for your question. This approach can be applied to other AM techniques. AM ontology is a kind of data structure to store domain knowledge. Accordingly, if a new process is added, classes related to the process can be extended to consider other AM processes, which are data structures for the new AM processes. Then, information on other AM processes can be stored in AM ontology. Once we construct an AM knowledge base for a target AM process, we can use the quantified relationship (a surrogate model) in the AM knowledge base to process optimization and local model modification procedure for the AM process. Thus, no significant change is required to use the proposed algorithm for other AM processes, but AM ontology construction and some experiments to quantify the relationship are required. 

We have added a paragraph on page 4: “If new AM technology is considered, such as Powder Bed Fusion and Direct Energy Deposition, AM ontology can embrace these AM processes by considering process parameters and material as classes of AM ontology.”

  1. The model is modified to reduce the overhang region, like figure 11, but what if in some cases it will change the mechanical performance of the model?
  • Thank you for your insightful question. Our research scope is design modification to avoid support structure while printing. We expected that the modification will not critically affect mechanical performance because the regions where modification is required are usually overhanging features. Accordingly, this paper less considered the relation between design modification and mechanical performance. We have been notified that model modification may lead to a change in mechanical performance. Accordingly, we have added new consideration of mechanical performance change as future work in Section 6: “Limitation of this study is that design modification may lead to mechanical performance change. Accordingly future study will consider how design modification affect me-chanical performance and consider more process parameters to ensure the relations design modification, process, and mechanical performance”.
  1. There are some typos in this manuscript, like section 3.3, "surrofate". Also there are a lot of sources not found in this document. The authors should proofread the manuscript to modify the typos and errors.
  • Thank you for your comment. We have thoroughly check typos and errors in the manuscript.

Reviewer 2 Report

The current manuscript presents a developed knowledge based design algorithm for support reduction of material extrusion additive manufacturing. Comprehensive and interesting data are presented, however, some major issues should be considered as follows:

- The abstract should be reduced and focused.

- The introduction section should be revised to be ended by the problem statement and the proposed solutions. Figure 1 and its description should be moved to the following section. 

- The manuscript organization should be completely edited. Design of the proposed algorithm, the material and the experimental setup should be presented in section 2.

- The results and discussion section should be existed, then sub-sections could be organized according to the authors' view. 

- The Figures should add more data or confirm the text data. Some figures do not add a meaningful data such as Figure 2.

- Some references are missing in the manuscript text, please revise this issue carefully.

- The number of references are not sufficient enough to justify the presented data. More of the recent studies should be cited.  

- The conclusion section should focus on the main contribution and novelty of the current work, it is recommended to use a bullet points style. 

In general, the presented work is valuable but still need some effort to be organized and revised according to the above comments and recommendations.  

Author Response

We appreciate your valuable comments on this manuscript. We have revised our manuscript to reflect the issues you pointed out. As a result, we believe the manuscript becomes more convincing with revision. We list our point-by-point answers to your questions in this rebuttal letter and have updated the manuscript based on the answers.

The current manuscript presents a developed knowledge-based design algorithm for support reduction of material extrusion additive manufacturing. Comprehensive and interesting data are presented, however, some major issues should be considered as follows:

- The abstract should be reduced and focused.

  • Thank you for your suggestion. We have revised the abstract with concise sentences as focusing our main logical flow, importance of support reduction – proposed algorithm – introduction of case studies.

- The introduction section should be revised to be ended by the problem statement and the proposed solutions. Figure 1 and its description should be moved to the following section.

  • We have updated introduction and Section 2 based on your comments. New sentences for problem statement are added in Introduction as below: “Research problem from literature review is that it is challenging to manage support structure generation during AM process, which leads to surface quality reduction and production time increment. Specifically, previous studies less considered how to avoid support structure generation in terms of AM process, part design, and quality of printed parts. Since support structure can be generated on the overhang feature due to the production manner of AM process, relationships among AM process, part design, and the quality of printed parts should be considered to ensure the quality and successful fabrication by AM.”

    Then, the proposed methods are described at the below of new sentences on page 3.

- The manuscript organization should be completely edited. Design of the proposed algorithm, the material and the experimental setup should be presented in section 2.

  • Thank you for your suggestion. We have re-titled Section 2 as ‘overview of knowledge-base design algorithm’ and described overall description of the proposed method. Accordingly, Figure 1 and corresponding descriptions are moved to Section 2.

- The results and discussion section should be existed, then sub-sections could be organized according to the authors' view.

  • Thank you for your comment. We have changed the title of Section 5 to “Result of case studies and discussion.” This section includes results from applications of the proposed algorithm. We also have tried to convey our observation and understandings.

- The Figures should add more data or confirm the text data. Some figures do not add a meaningful data such as Figure 2.

  • Information in Figure 2 is very important for developing AM ontology, which is fundamental structure of the knowledge base in this study. Tables and figures in Section 3.1 were provided to specify the relations in Figure 2. Based on your comment, we have improved Figure 2. Also, we have added more description for Figure 2, which is “Geometry, process, and physics are closely related to support structure generation and are critical factors that influence quality of final product printed by AM. These relations are key information for developing knowledge base that can be used to infer knowledge that users want to know.”

- Some references are missing in the manuscript text, please revise this issue carefully.

  • Thank you for your comment. We have used software for managing references, but found out missing information. For example, a typo in reference 3 is ‘echanical engineering’ and dash(-) on title of reference 5 was represented as ‘&mdash’. We carefully double checked the reference and amended these issues.

- The number of references are not sufficient enough to justify the presented data. More of the recent studies should be cited.

  • We have added 3 more references related to the relations between MEX process parameters, geometry, and the quality on the sentence above Figure 2. New references are also added in the reference section as 22, 23, and 24.

- The conclusion section should focus on the main contribution and novelty of the current work, it is recommended to use a bullet points style.

  • We have revised conclusion section as you recommended. Newly added sentences are as below:
    “ The contribution and novelty of the proposed methods are described as below:
    • Knowledge base development to formalize and store relations among geometry, process, and quality
    • Knowledge base application to retrieve predominant process parameters related to surface roughness
    • Automatic design modification by selecting and redesigning specific overhang regions to avoid support structure generation and ensure surface quality, which is an objective function of support reduction algorithm
    • Process optimization to avoid support structure generation of the selected specific overhang regions “

In general, the presented work is valuable but still need some effort to be organized and revised according to the above comments and recommendations.

Round 2

Reviewer 2 Report

The revised manuscript is improved, however, some issues still need to be considered as follows:

- In Figures 3,4 and 5, the font is too small to be readable, please try to focus on the main data to be presented.

- The standard deviation should be added to the plots in Figure 7.

- There should be a section of materials and experimental tests including the following:

      - Case study material properties and process parameters

      - Number of samples

      - Instruments and experimental procedures 

      - number of measurements for each sample and the standard deviation          for surface roughness

- What about the dimensional accuracy for parts produced with proposed technique?  

The presented work is very interesting but another review is highly recommended.

Author Response

We appreciate your valuable comments on this manuscript. We have revised the manuscript based on your comments. We answer your questions point-by-point and we have updated the manuscript based on the answers. Answers to the questions are represented with ‘italic’ font, while the questions are represented with bold font in this letter.

The revised manuscript is improved, however, some issues still need to be considered as follows:

Q1. In Figures 3,4 and 5, the font is too small to be readable, please try to focus on the main data to be presented.

 - Figures 3, 4, and 5 are captured from the Protégé software. To ensure legibility, we have enlarged the figures and increased their resolution.

- We aim to show class hierarchy in Figure 3. However, the figure presents repeated information. Thus, we have removed a part of Figure 3, Figures 3 (a) and (b), and we have enlarged the Figure 3 (c) for legibility.

 - We have changed the resolution of Figure 4.

 - We leave the figure without modification because it is important to show implementation in the software. However, we have enlarged the size of Figure 5 for legibility.

Q2. The standard deviation should be added to the plots in Figure 7.

- We have inserted the error bar of overall data into the plots in Figure 7.

Q3. There should be a section of materials and experimental tests including the following:

- Section 3.3 presents the materials and experimental settings for printing coupons for parametric studies.

    - Case study material properties and process parameters

- For all fabrication and experiments, polylactic acid (PLA) filaments from Esun® and preset process parameter settings in Table 4 are applied. Process parameters (layer thickness and deposition speed) related to the parametric study and case study are changed according to the cases. We cannot represent details of the optimized deposition speed because the optimized speed is different for each deposition path.

- We added the filament information in Section 3.3.

      - Number of samples

- We have described the number of samples in Section 3.3 on page 9. The combination of selected three process parameters yields 30 different process parameters setting. Accordingly, we printed 90 study coupons for the parametric study.

      - Instruments and experimental procedures 

- The picture of in-house MEX printer is added to Figure 6 (b). We use the printer for fabrication of parts for parametric and case study. The surface roughness for all coupons is measured in Dektak® XT-E stylus profiler.

- We also have modified paragraphs in Section 3.3 to describe experimental procedure.

      - Number of measurements for each sample and the standard deviation for surface roughness

- We made one measurement per a coupon. (Added to Section 3.3)

- We have added the standard deviation to Table 5 in Section 3.3

Q4. What about the dimensional accuracy for parts produced with proposed technique?  

- In this research, the dimensional accuracy is not the primary performance index of the proposed algorithm because the presented approach utilizes the local geometry modification algorithm, which leads to changes in the dimension compared with original geometries. Furthermore, parts with complex geometries are difficult to check their dimensional accuracy, specifically complex curved areas. Thus, we do not include observation of dimensional accuracy because presenting dimensional accuracy is not meaningful, but it can cause confusion for readers.

- We have not modified the manuscript for this question. However, since dimensional accuracy is a factor related to the quality of the final part, we have added a sentence in Conclusion as a future work as followings: “Accordingly, future study will consider how design modification affect the quality of final parts including mechanical performance and dimensional accuracy that are not concerned in this study.”

The presented work is very interesting but another review is highly recommended.

- The end -

Round 3

Reviewer 2 Report

The revised manuscript is significantly improved, and the review comments and recommendations are well addressed. As a minor point, the data regarding the sample preparation, experimental tests, and materials material should be included in a separate section titled "Materials and methods".